# Can Menzerath's law be a criterion of complexity in communication?

Iván G. Torre[1☺], Łukasz Dębowski[2☺], Antoni Hernández-Fernández[3,4]*

**1** Vicomtech Foundation, Basque Research and Technology Alliance (BRTA), San Sebastián, Spain, **2** Institute of Computer Science, Polish Academy of Sciences, Warszawa, Poland, **3** Complexity and Quantitative Linguistics Lab, Institut de Ciències de l'Educació, Universitat Politècnica de Catalunya, Barcelona, Catalonia, Spain, **4** Societat Catalana de Tecnologia, Secció de Ciències i Tecnologia, Institut d'Estudis Catalans, Barcelona, Catalonia, Spain

☺ These authors contributed equally to this work.
* antonio.hernandez@upc.edu

**Data Availability Statement:** The data are held in a public repository with URL: https://github.com/ivangtorre/Can-Menzerath-law-be-a-criterion-of-complexity-in-communication.

## Abstract

Menzerath's law is a quantitative linguistic law which states that, on average, the longer is a linguistic construct, the shorter are its constituents. In contrast, Menzerath-Altmann's law (MAL) is a precise mathematical power-law-exponential formula which expresses the expected length of the linguistic construct conditioned on the number of its constituents. In this paper, we investigate the anatomy of MAL for constructs being word tokens and constituents being syllables, measuring its length in graphemes. First, we derive the exact form of MAL for texts generated by the memoryless source with three emitted symbols, which can be interpreted as a *monkey typing* model or a null model. We show that this null model complies with Menzerath's law, revealing that Menzerath's law itself can hardly be a criterion of complexity in communication. This observation does not apply to the more precise Menzerath-Altmann's law, which predicts an inverted regime for sufficiently range constructs, i.e., the longer is a word, the longer are its syllables. To support this claim, we analyze MAL on data from 21 languages, consisting of texts from the Standardized Project Gutenberg. We show the presence of the inverted regime, not exhibited by the null model, and we demonstrate robustness of our results. We also report the complicated distribution of syllable sizes with respect to their position in the word, which might be related with the emerging MAL. Altogether, our results indicate that Menzerath's law—in terms of correlations—is a spurious observation, while complex patterns and efficiency dynamics should be rather attributed to specific forms of Menzerath-Altmann's law.

## 1 Introduction

Identification and characterization of universal statistical regularities present in human and non-human communication is the ultimate goal of quantitative linguistics, linguistic laws probably being its major exponent. In this paper we will focus on one quantitative linguistic law that has different flavors: Menzerath's law and Menzerath-Altmann's law. There was a little known precedent [1, 2] before Paul Menzerath experimentally observed that, on average, the

**Funding:** This work has been funded by the project PRO2021-S03-HERNANDEZ (Institut d'Estudis Catalans), where AHF is the principal investigator. URL: https://futur.upc.edu/30546321 AHF is also funded by the grant TIN2017-89244-R from Ministerio de Economia, Industria y Competitividad (Gobierno de España) and supported by the recognition 2017SGR-856 (MACDA) from AGAUR (Generalitat de Catalunya). URL: https://futur.upc.edu/22024384 The funders had no role in study design, data collection and analysis, decision to publish, or preparation of the manuscript.

**Competing interests:** The authors have declared that no competing interests exist.

longer—in the number of syllables—a word was, the shorter—in seconds—those syllables were [3, 4]. This empirical observation, was verified in multiple languages and qualitatively generalized as *the larger the size m of a linguistic element is, the smaller the average size y of its constituents is* and became known as *Menzerath's law* [5]. Later, Gabriel Altmann stated this observation as a more precise mathematical formula that was called thereafter *Menzerath-Altmann's law* (MAL) [6–8]:

$$y = \alpha m^{\beta} \exp\left(-\gamma m\right) \qquad (1)$$

where $\alpha, \beta, \gamma$ are empirical parameters, being $\alpha > 0$, and usually $\beta < 0$ and $\gamma < 0$.

MAL has been explained as a manifestation of complex behavior [9], although the interpretation of parameters $\alpha$, $\beta$, and $\gamma$ remains unknown [10], except for some basic correspondences with the description of mean length of linguistic elements [11]. Recently, it was noticed [12] that when the product of parameters $\beta$ and $\gamma$ is positive, $\beta \cdot \gamma > 0$, then Eq (1) has an extremum at $n^* = \beta/\gamma$, leading in this case to a second regime in MAL where *the larger the whole, the larger the size of its constituents*, fairly against the well-known qualitative version of Menzerath's law. Actually Menzerath had observed this fact [3, 4] but this second regime was little explored later, if not ignored [12]. In this paper, we will explore and distinguish between (i) *Menzerath-Altmann's law (MAL)*, when data is fitted to Eq (1) and (ii) *Menzerath's law*, when the analysis is reduced to the study of correlations, between the size of the word in number of syllables and the size of the syllables in graphemes.

MAL and Menzerath's law have intensely been researched within the field of quantitative linguistics including written corpora [10, 13, 14], speech [12, 15], and even music [16]. Besides, similar analyses can be extended to any system that forms a hierarchy of units of different levels. Consequently, Menzerath's law has also been observed in genomes [17–19], protein domains [20], penguin vocalizations [21], chimpanzee gestural communication [22] and primate vocalizations [23–27], including strepsirrhine species [28]. It is related to compression principles [23, 29] and explained in terms of oral communication as follows: Given that lungs' air capacity is limited, when longer words are vocalized then their constituents tend to be more compressed because the speaker must 'hurry' to finish articulating the word before running out of air [12, 25].

However, some authors have discussed whether MAL or Menzerath's law could be trivially explained in particular cases [19, 30] or sometimes, just quietly not be observed [26]. Nonetheless, the lack of a standardized methodology of studying these statistical laws introduces a large variation in the results depending on the segmentation method selected, the units of study, and some additional variables that we will address here [10, 14, 31]. Furthermore, to fully consider whether MAL and Menzerath's law have an explanation from the point of view of compression and emerging complexity, we will do the exercise of studying them for so-called *monkey typing* models [32], which have previously heated the debate about other linguistic laws such as Zipf's law and brought stimulating results to the scientific community [33, 34].

It is worth addressing that linguistic structures present in written texts are probably a reflection of orality, although some complex structures in writing may be rare in orality. The rich repertoire of speech sounds in human communication is produced by modifying the exhausted air throughout the speech apparatus and can be first categorized in two types of sounds that have their correspondence on graphemes in alphabetic scripts: vowels and consonants [35]. Vowels are produced when pulmonary air passes unhindered through the upper vocal tract, whereas in the case of consonants there is always some kind of obstruction above the level of the larynx [36, 37]. Although its conception has evolved extensively throughout history, syllable is considered a linguistic structure made up of a sound sequence with a *nucleus*

(commonly a vowel or a syllabic consonant) that can be preceded by an *onset* (a consonant or a consonant cluster) and/or followed by a *coda* (frequently a consonant). In this paper, we will adopt word tokens, syllables and graphemes as the linguistic units to massively study, in written texts, whether MAL and Menzerath's law are universal non-spurious observations or not.

The contents of this paper are as follows. First, we will derive an exact equation for Menzerath's law for a memoryless source that emits three distinct symbols (the *monkey typing* model in the sense of [32]), which we will later use as a baseline to compare with experimental data. In order to explore MAL in written texts in a systematic way, we will syllabify and analyze 21 different languages of the Standardized Gutenberg Corpus [38]. We will report that different methods of computing the average syllable length yield similar parameters of MAL, which supports some universality of the law. Then, we will show that only some languages seem to comply with Menzerath's law, while a simple null model can mimicry this statistical law. However, this memoryless source model cannot reproduce the inverted regime of MAL. Subsequently, we will seek for the origins of MAL in the empirical distribution of the syllable size relative to its position in the word, which is investigated for the first time. We finally address the significance of our results to explain if Menzerath's law and MAL can be a criterion of complexity in communication.

## 2 Menzerath-Altmann's law and null models

The idea that simple stochastic processes producing random sequences of characters, phonemes, or other linguistic units may mimic quantitative linguistic laws has been long addressed in the literature and served as a starting point to distinguish complex dynamics from spurious observations [32, 33, 39]. However, such a discussion has mostly focused on the study of Zipf's law, where it has been shown that texts generated by memoryless sources, graphically called *monkey typing*, exhibit a version of this law [32, 40–42]. Only later it has been contested that those results differ substantially from Zipf's law for natural language [34].

Random models for Menzerath-Altmann's law were previously discussed in Linguistics at the sentence-clause level considering several randomized versions of the original text [43], and also for word length motifs on an extensive Modern Greek corpus [44], where the inverted regime in MAL was not considered by the random model (see Fig 1 in [44]). Recently, Menzerath's law has been recovered in the syntax of several languages with random sentences, determining with some approximation that this law can be reproduced at the syntactic level by randomization [45].

In the context of a study of chromosomes, Menzerath's law was claimed a spurious observation by applying a simple stochastic model [30] but later this claim was contested on account of ignoring well-formedness of chromosomes [19]. In this section, we are going to consider emergence of Menzerath's law for a simple memoryless source, i.e., a *monkey typing* model. Similar analyses could be extended to more complicated stochastic models such as finite-state sources, i.e., hidden Markov models. Such models can help us to understand whether Menzerath's law is a spurious observation or which aspects of the more specific Menzerath-Altmann law may be considered symptoms of a complex behavior.

Our first observation is that Eq (1) can be specialized for texts generated by stochastic processes as follows: Let $N$ and $M$ be the numbers of consonants and vowels, respectively, in a randomly generated word. Then, the mean length of a syllable equals $\frac{N+M}{M}$, if we assume that the number of syllables in the word can be approximated as the number of vowels $M$. In particular, Menzerath-Altmann's law states that the conditional expectation $E\left(\frac{N+M}{M}\big|M=m\right)$ is a

decreasing function of number of syllables $m$ of the particular form

$$E\left(\frac{N+M}{M}\Big|M=m\right) = \alpha m^{\beta} \exp\left(-\gamma m\right) \tag{2}$$

with three parameters: $\alpha$, $\beta$, and $\gamma$. Although this is the most commonly accepted form of stating Menzerath-Altmman's law, there are other equivalent proposals presupposing two parameters [12] as well as other non-equivalent forms [6].

Our second observation is that in order to specify random variables $N$ and $M$, it suffices to consider stochastic processes that produce exactly three distinct symbols: symbol $C$ (consonant), symbol $V$ (vowel) and symbol $S$ (word delimiter). The respective stochastic source runs in an infinite loop producing random texts of an arbitrary length, i.e., random sequences of $C$'s, $V$'s, and $S$'s. Formally, for any stochastic process of this kind, we can treat the strings of $C$'s and $V$'s as words separated by $S$'s and we can investigate mathematically the expected number $N$ of consonants in a random 'word' containing $M$ vowels. Such models can provide a highly demanded theoretical underpinning for the Menzerath-Altmann's law and can possibly explain its empirical parameters in terms of the detailed construction of a generating stochastic process. In the following, we will consider the simplest model that reproduces some form of Menzerath's law, which is the memoryless source.

## 2.1 The exact form of Menzerath's law for the memoryless source

The memoryless source emits a sequence of independent identically distributed random variables. For this reason, it has been sometimes graphically called *monkey typing* in the quantitative linguistic literature [32, 33]. Formally, it can be represented as a hidden Markov model, i.e., a finite-state source, that has only one state and emits in an infinite loop symbols $C$, $V$, and $S$ with probabilities $p_c$, $p_v$, and $1 - p_c - p_v$, respectively—see Fig 1. Let us restate the definition our target, being two random variables:

- $N$—the number of $C$'s generated between two $S$'s,

- $M$—the number of $V$'s generated between two $S$'s.

First, we are interested in computing conditional expectation $E(N|M = m)$. In fact, the derivation of this conditional expectation can be substantially simplified when we introduce random variables:

- $N_i$—the number of consecutive $C$'s generated directly after the $i$-th $V$ (or after $S$ in case of $i = 0$).

Since $N = \sum_{i=0}^{M} N_i$ and random variables $N_i$ are identically distributed as $P(N_0 = n) = p_c^n(1 - p_c)$ and independent of $M$, hence we may write:

$$\begin{aligned}
E(N|M = m) &= \sum_{i=0}^{m} E(N_i|M = m) = \sum_{i=0}^{m} EN_i = (m+1)EN_0 \\
&= (m+1)\sum_{n=0}^{\infty} nP(N_0 = n) = (m+1)\sum_{n=0}^{\infty} np_c^n(1 - p_c) \\
&= \frac{p_c}{1 - p_c}(m+1).
\end{aligned} \tag{3}$$

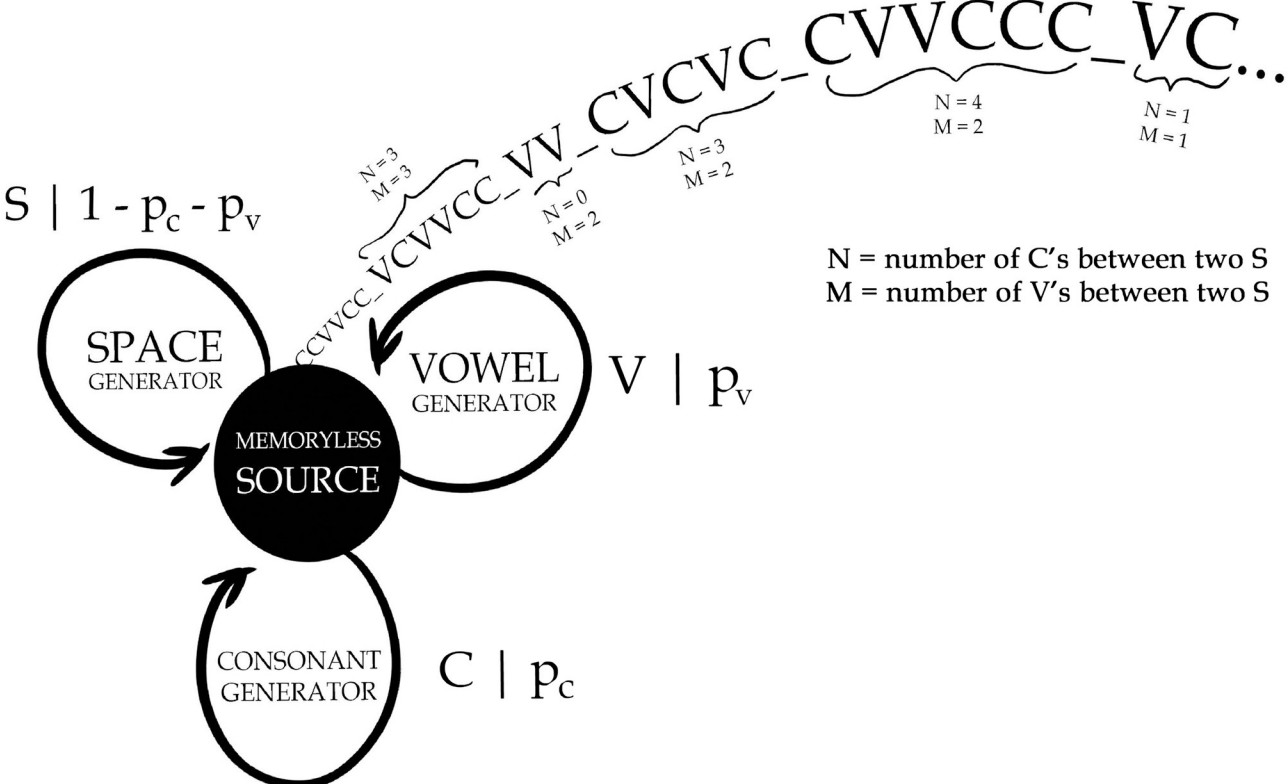

**Fig 1. Memoryless source model.** The memoryless source has only one state and emits in an infinite loop symbols *C*, *V* and *S* with probabilities $p_c$, $p_v$, and $1 - p_c - p_v$, respectively.

As a result we obtain formula

$$E\left(\frac{N+M}{M}|M=m\right) = \frac{a}{m} + b,$$ (4)

where

$$a = \frac{p_c}{1-p_c}, \quad b = 1 + \frac{p_c}{1-p_c}.$$ (5)

Conceptually, the memoryless source is the simplest possible stochastic process to make Menzerath's law observable. The respective regression formula differs from Menzerath-Altmann's law (2) and has only one degree of freedom, since parameters *a* and *b* depend only on parameter $p_c$. The memoryless source is a baseline model to be considered before proposing more complicated hidden Markov models. Let us observe that some limitation of the above model is that the nucleus of the syllable is always formed by a single vowel. However, in natural language, the syllabic nucleus can also be written by two or three graphemes yielding diphthongs or triphthongs [46, 47]. We will evaluate the validity of this simplification depending on the particular human language.

## 2.2 Numerical simulations and validity

As a sanity check to our mathematical derivations, we will demonstrate the validity of formula (4) on a simulation of random texts generated by the memoryless source from Fig 1. For

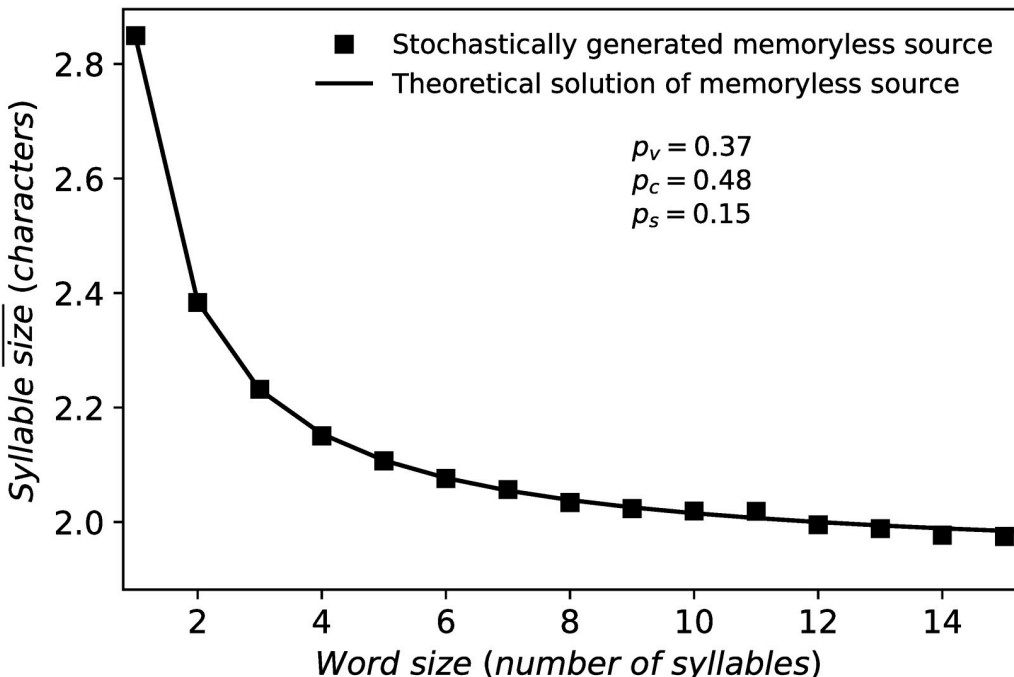

**Fig 2. Menzerath-Altmann's law for the memoryless source.** Parameter $p_c = 0.48$ is used accordingly with real languages. The solid line is the theoretical model, given by Eqs (4) and (5), whereas the square points are the simulation of the memoryless source.

parameters $p_c = 0.48$ and $p_v = 0.37$, which approximate the relative frequencies of consonants and vowels for human languages, we have generated a random text of length $10^6$. Then we have computed the empirical averages of statistic $\frac{N+M}{M}$ conditioned on event $M = m$, discarding all cases when the count of event $M = m$ was less than 10. The results of our simulations are plotted in Fig 2, where we can observe a very good agreement between the theory and the numerical simulation.

The special form (4) for the memoryless source originates from constancy of conditional expectations $E(N_i|M = m)$ with respect to position $i$, that is, each consonant cluster in a word is equally long on the average. In contrast, for natural language conditional expectations $E(N_i|M = m)$ depend both on $i$ and $m$ in quite a complex way, as we will exhibit later in this paper. As it can be seen in Fig 2, the memoryless source reproduces Menzerath's law in the form *the larger the linguistic construct, the smaller the size of its constituents*. It fails to reproduce the inverted regime, where the average length of the constituents grows with the length of the construct.

## 3 Material and methods

### 3.1 Gutenberg Corpus

For this work, we have used the Standardized Gutenberg Corpus database which constitutes a curated standardized open access version of the complete Project Gutenberg containing more than 50000 written books in several languages [38]. For our analysis we consider only books written in one language (monolingual texts) out of 28 languages that entail 5 or more monolingual books. Additionally, we have set an upper limit of 2500 randomly selected books for the

**Table 1. The adopted grapheme classification into seven sonority classes.**

| Sonority class | Graphemes |
|---|---|
| Vowels | a, e, i, o, u, y, à, á, â, ä, æ, ã, å, ã, ą, è, é, ê, ë, ē, ė, ę, î, ï, í, ī, į, ì, ô, ö, ò, ó, œ, ø, ō, õ, û, ü, ù, ú, ū, ů, ÿ, ű, ő, ŵ, ŷ, ỳ, ẁ, ě, ý, ǫ, |
| Approximants | ŭ, w, ł |
| Liquid | l, r, ř |
| Nasals | m, n, ñ, ń, ŋ, ň, |
| Fricatives | ß, z, v, s, f, ç, ć, ś, ŝ, ĉ, ĥ, h, ĵ, š, ž ð, đ |
| Affricates | x, j, ź, ż, ĝ, č |
| Occlusives | b, c, d, g, t, k, p, q, þ, ď, ť |

case of languages exceeding that value. This was the case of French, with almost 3000 available books and English with more than 45000. Moreover, we additionally filtered out 7 languages not based on the Latin alphabet, as they would need additional considerations on the automatic syllabification process (see section 3.2). Then, within each book we have filtered out all words containing characters not included in Table 1. The percentage of thus excluded words less amounted than 0.01% and included mostly loanwords.

In conclusion, we apply the analysis to a total of 21 languages from the Standardized Gutenberg Corpus, resumed in Table 2 and encompassing more than 10000 books and roughly $6 \cdot 10^8$ words tokens.

**Table 2. Summary of characteristics for the 21 languages studied including the total number of books, word tokens, syllables, consonants (*C*) and vowels (*V*), as well as the probabilities of vowels (*$p_v$*), consonants (*$p_c$*) or spaces (*$p_s$*).** The last column (*Syll/V*) reports the syllable-vowel ratio.

| Language | Books | Tokens | Syll. | C | V | $p_v$ | $p_c$ | $p_s$ | Syll/V |
|---|---|---|---|---|---|---|---|---|---|
| English (en) | 2500 | $1.5 \cdot 10^8$ | $2.4 \cdot 10^8$ | $4.0 \cdot 10^8$ | $2.6 \cdot 10^8$ | 0.33 | 0.49 | 0.18 | 0.89 |
| French (fr) | 2500 | $1.5 \cdot 10^8$ | $2.5 \cdot 10^8$ | $3.7 \cdot 10^8$ | $3.0 \cdot 10^8$ | 0.37 | 0.45 | 0.18 | 0.82 |
| Finnish (fi) | 2226 | $7.7 \cdot 10^7$ | $1.9 \cdot 10^8$ | $2.6 \cdot 10^8$ | $2.4 \cdot 10^8$ | 0.42 | 0.45 | 0.13 | 0.80 |
| German (de) | 1801 | $8.5 \cdot 10^7$ | $1.5 \cdot 10^8$ | $2.8 \cdot 10^8$ | $1.7 \cdot 10^8$ | 0.32 | 0.52 | 0.16 | 0.87 |
| Italian (it) | 834 | $5.1 \cdot 10^7$ | $1.0 \cdot 10^8$ | $1.3 \cdot 10^8$ | $1.2 \cdot 10^8$ | 0.39 | 0.44 | 0.17 | 0.91 |
| Dutch (nl) | 817 | $4.5 \cdot 10^7$ | $7.3 \cdot 10^7$ | $1.3 \cdot 10^8$ | $8.9 \cdot 10^7$ | 0.34 | 0.49 | 0.17 | 0.82 |
| Spanish (es) | 650 | $4.2 \cdot 10^7$ | $8.1 \cdot 10^7$ | $1.0 \cdot 10^8$ | $9.0 \cdot 10^7$ | 0.39 | 0.43 | 0.18 | 0.89 |
| Portuguese (pt) | 565 | $1.5 \cdot 10^7$ | $2.9 \cdot 10^7$ | $3.6 \cdot 10^7$ | $3.3 \cdot 10^7$ | 0.39 | 0.43 | 0.18 | 0.87 |
| Hungarian (hu) | 237 | $1.2 \cdot 10^7$ | $2.6 \cdot 10^7$ | $3.8 \cdot 10^7$ | $2.7 \cdot 10^7$ | 0.35 | 0.49 | 0.16 | 0.96 |
| Swedish (sv) | 206 | $8.6 \cdot 10^6$ | $1.4 \cdot 10^7$ | $2.6 \cdot 10^7$ | $1.5 \cdot 10^7$ | 0.30 | 0.52 | 0.18 | 0.99 |
| Esperanto (eo) | 106 | $2.2 \cdot 10^6$ | $4.1 \cdot 10^6$ | $5.8 \cdot 10^6$ | $4.4 \cdot 10^6$ | 0.35 | 0.47 | 0.18 | 0.94 |
| Latin (la) | 88 | $3.9 \cdot 10^6$ | $8.4 \cdot 10^6$ | $1.2 \cdot 10^7$ | $9.4 \cdot 10^6$ | 0.48 | 0.47 | 0.15 | 0.89 |
| Danish (da) | 70 | $3.9 \cdot 10^6$ | $6.4 \cdot 10^6$ | $1.1 \cdot 10^7$ | $6.7 \cdot 10^6$ | 0.31 | 0.50 | 0.19 | 0.96 |
| Tagalog (tl) | 57 | $9.9 \cdot 10^5$ | $2.0 \cdot 10^6$ | $2.7 \cdot 10^6$ | $2.3 \cdot 10^6$ | 0.38 | 0.45 | 0.17 | 0.90 |
| Catalan (ca) | 32 | $1.2 \cdot 10^6$ | $2.0 \cdot 10^6$ | $2.8 \cdot 10^6$ | $2.2 \cdot 10^6$ | 0.36 | 0.45 | 0.19 | 0.89 |
| Polish (pl) | 29 | $4.1 \cdot 10^5$ | $8.3 \cdot 10^5$ | $1.2 \cdot 10^6$ | $9.2 \cdot 10^5$ | 0.35 | 0.49 | 0.16 | 0.90 |
| Norwegian (no) | 20 | $8.0 \cdot 10^5$ | $1.2 \cdot 10^6$ | $2.1 \cdot 10^6$ | $1.3 \cdot 10^6$ | 0.31 | 0.50 | 0.19 | 0.95 |
| Czech (cs) | 10 | $3.6 \cdot 10^5$ | $7.2 \cdot 10^5$ | $1.0 \cdot 10^6$ | $7.2 \cdot 10^5$ | 0.34 | 0.48 | 0.17 | 0.99 |
| Welsh (cy) | 10 | $2.2 \cdot 10^5$ | $3.3 \cdot 10^5$ | $5.6 \cdot 10^5$ | $3.7 \cdot 10^5$ | 0.33 | 0.48 | 0.19 | 0.88 |
| Icelandic (is) | 7 | $7.9 \cdot 10^4$ | $1.3 \cdot 10^5$ | $2.1 \cdot 10^5$ | $1.4 \cdot 10^5$ | 0.32 | 0.50 | 0.18 | 0.96 |
| Afrikaans (af) | 6 | $2.0 \cdot 10^5$ | $3.0 \cdot 10^5$ | $4.9 \cdot 10^5$ | $3.8 \cdot 10^5$ | 0.36 | 0.46 | 0.18 | 0.77 |

## 3.2 Syllabification

Syllabification, which means the division of a word into syllables, is an intricate procedure whose exact algorithm remarkably differs depending on the language considered [36, 48–51], hindering the process to reach a common automatic method that may be applied to multilingual corpora. The variety of approaches to the automatic syllabification procedure includes: dictionary-based look-up procedures or rule-based systems [52, 53]; the Hyphenation methods, based on Hunspell hyphenation dictionaries (e.g. such those used for *LibreOffice* or *OpenOffice* spell checkers) [54, 55]; the Legality Principle, based on the Maximal Onset Principle, which determines the tendency of the consonants to be part of the onset instead of the syllable coda [49, 50]; or data-driven trained neural networks [56, 57], among some other methods.

For this analysis, we consider the Sonority Sequencing Principle (SSP) as the most suitable option because of its adaptability to multilingual domains, accuracy, and efficiency [56]. SSP states that the nucleus of the syllable is the element of the maximum sonority. Although some linguistic varieties of languages not studied here (Russian, Arabic) may not strictly follow this principle and various formulations of SSP have been discussed, the SSP states that only sounds of higher sonority rank are permitted between any element of a syllable and the syllable peak, considering that clusters violating this principle do occur but they are infrequent ([58], p.285).

The sonority classification depends on the type of sound, usually following a hierarchy considered universal by some authors but it can differ to some extent from language to language [59, 60]. For Romance languages it is considered that the sonority scale ranges from the most open sounds of vowels to the most closed sounds of occlusive consonants, according to the following gradation [49, 51, 61]:

$$\text{vowels} > \text{approximants} > \text{liquids} > \text{nasals} > \text{fricatives} > \text{affricates} > \text{occlusives}$$

However, sonority classes are still under some discussion [62] and consequently SSP has some limitations (see [49] for a review), including that each character corresponds to a single sonority class, which is not always true [63]. For instance, in Spanish grapheme *y* can behave as a vowel in the conjunction *y* (meaning *and*) or as a consonant *y* in the word *yate* (*yacht*). In the case of an ambiguous sonority class, we have assigned graphemes to their most frequent sonority class as summarized in Table 1. Assuming this sonority classification, the syllabification process has been carried out automatically implementing the SSP gradation within the NLTK Python package [64].

## 3.3 Statistical procedures

In order to check Menzerath's law we use Spearman's rank correlation test to determine if there is a monotonic function that explains the relation between the sizes of words in the number of syllables and the mean size of syllables. It is computed for all languages, both the correlation coefficient that may range from −1 to +1, and the associated p-value to test the level of significance against the null hypothesis. The exact regression (4) for the memoryless source is a monotonically decreasing function, so Spearman's rank correlation coefficient for the memoryless source equals −1.

Additionally, a lower threshold has been set to filter out word lengths that appear less than 25 times, as those may be considered outliers without statistical value. Note that this filter may affect slightly differently to the two procedures discussed on section 4.2. Finally, parameter fitting has been addressed using the Levenberg–Marquardt algorithm and the goodness of fit has been determined by computing the coefficient of determination $R^2$.

Reproducibility of the paper is addressed in Section 7. We provide full access to data, scripts and algorithms used within this research.

## 4 Results

### 4.1 Analysis of the Standardized Gutenberg corpus

To test the validity of Menzerath's law, Menzerath-Altmann's law, and the memoryless source model, graphemes are clustered into three types of symbols: consonants (*C*), vowels (*V*) and spaces (*S*). The graphemes that belongs to the group of vowels are summarized in the first row of Table 1, the group of consonants include any grapheme classified as approximants, liquids, nasals, fricatives, affricatives and occlusives, while spaces include any punctuation symbol that separates words. Considering this classification, the number of spaces in a given text is exactly one unit less than the number of words, so we approximate here, without loss of generality, that the total number of spaces in a text is equal to the number of tokens.

Table 2 summarizes, for each language, the main characteristics of the corpus, including total number of books, word tokens and syllables, total number of *C*'s and *V*'s, as well as the probability of a vowel ($p_v$), a consonant ($p_c$), and a space ($p_s$). Besides, in the last column, the syllable-vowel ratio (*Syll*/*V*) is shown, which serves us to test the validity of the hypothesis, assumed in the memoryless source model, that syllables are approximately composed of one vowel. *Syll*/*V* reflects that this assumption is mostly fulfilled for languages like Hungarian, Swedish, Danish, or Czech with values greater that 0.95, while can not be generalized across all languages, as for example French, Finnish, Dutch, or Afrikaans reveal values less than 0.85.

### 4.2 Menzerath-Altmann's law for full corpus and averaged by book

The first question to be considered is what is the most appropriate procedure to calculate Menzerath-Altmann's law once the corpus and linguistic levels of study have been identified. While for other quantitative linguistic laws, Zipf's law in particular, previous research addressed extensive discussions on the statistical methodologies, null models, finite size effects and spurious artifacts [65–67], this is not the case of MAL. Some discussion of the null models is considered in the next section. Here we analyze the difference between two distinct approaches to fit MAL: (i) computing MAL for the full corpus considered as a single text and (ii) computing MAL for each book and later obtaining the average. It is not guaranteed that these two procedures are equivalent. For the sake of clarity we propose an extreme case where there are two books: the first one has one hundred words of one syllable, being all of them one character long, whereas the second book has only one word of one syllable, but this syllable is two characters long. The first procedure of computing MAL would reach that the mean size of syllables with one word is approximately 1 character long while the second procedure would result in the mean size being 1.5 characters long.

We apply both methods to calculate the relation between the word size measured in number of syllables versus the mean size of those syllables for all languages reported in Table 2. The results for English, French, Finnish and German are presented in the panels of Fig 3, while the results for the rest of languages are depicted in S1 Fig. Each thin grey line represents one book, and the average over them is displayed using black circles joined together for visual representation. Blue square points represent the MAL computed directly on the full corpus treated as a single text. While at the level of texts, a notable variability commonly found in linguistic laws [67] is visible, there are hardly any differences between both methods on the level of the full corpus. In the following, Menzerath's law, and Menzerath-Altmann's law will be addressed and fitted only considering the second method.

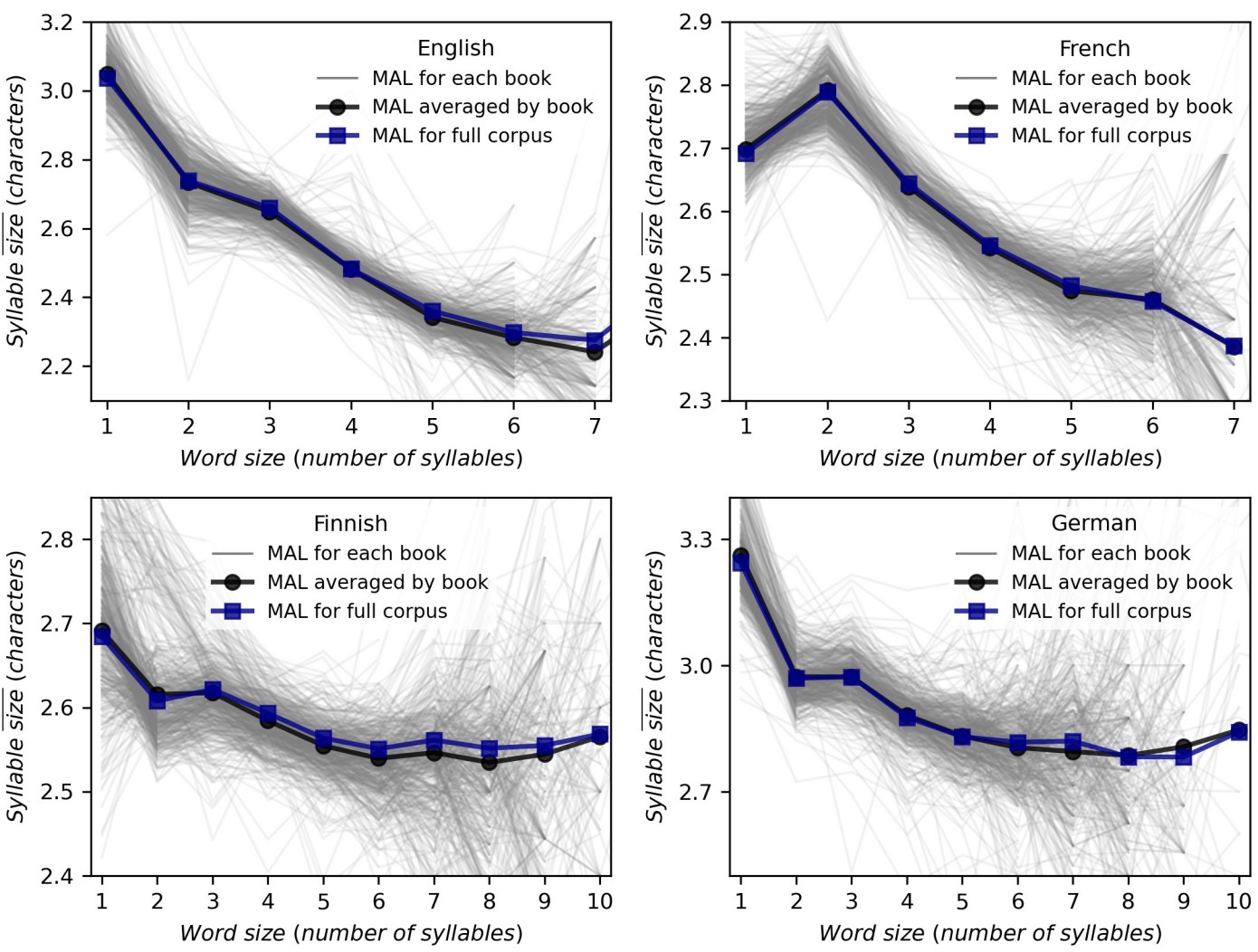

**Fig 3. Different methods of computing MAL reflect similar results.** Relation between word size (measured in number of syllables) versus the mean size of those syllables, where each panel corresponds to one different languages of Gutenberg corpus. Each thin grey line represents one book, whereas black circles are the mean duration of books. Meanwhile blue squares are the result of computing MAL from the full corpus. Both methods shows similar results and solid lines are just represented for visual comparison. Results on 17 additional languages are provided in S1 Fig.

## 4.3 Menzerath's law correlation results

To test whether there is a negative correlation between the size of the word and the mean size of its syllables we tested Spearman's rank correlation on the full corpus. Given that constituents with one part frequently do not fulfill the law [12], we remove them from the analysis and consider only words with two or more syllables (Fig 4 and S2 Fig), although analyses including words with one syllable are also reported in S3 Fig. Results are summarized in Table 3, showing that approximately a half of the considered languages supports statistically significant evidences of satisfying Menzerath's law with p-value <0.05 and a half does not. On the other hand, the theoretical solution for the memoryless source in Eq (4) corresponds to a monotonically decreasing function between the construct size and the mean size of the parts, so its Spearman's rank correlation is −1, fully complying with Menzerath's law. While more inquiries could be done to test whether the failure of Menzerath's law for some languages can be

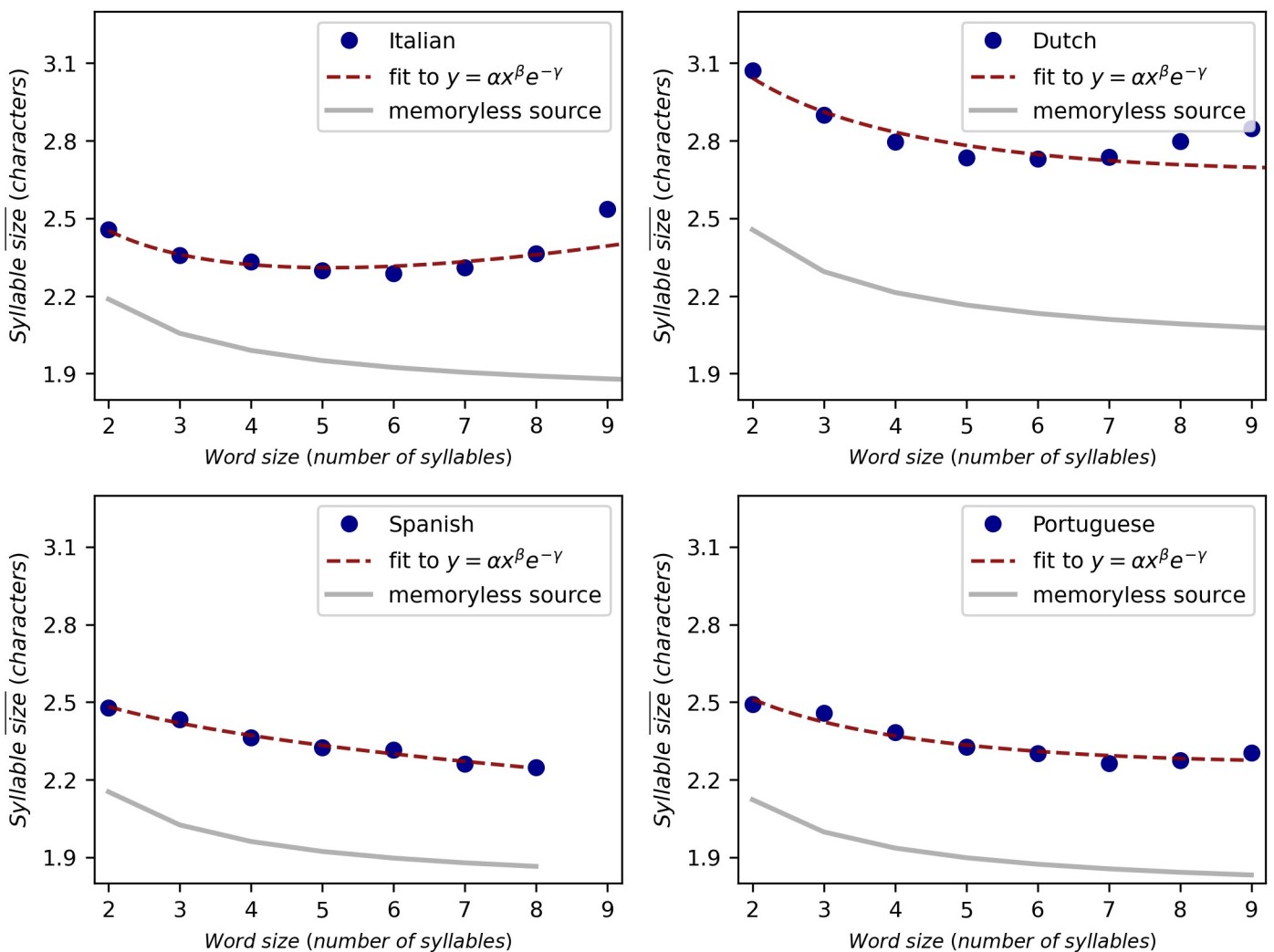

**Fig 4. Menzerath-Altmann's law and the memoryless source baseline.** Relation between the word size and the mean size of syllables for Italian, Dutch, Spanish and Portuguese. Experimental results are shown in blue circles, the red dotted line is a fit to Menzerath-Altmann's law (2), whereas the gray solid line corresponds to Eq (4) with $p_c$ given by the relative frequency of consonants. Results for 17 additional languages are provided in S2 Fig.

due to statistical artifacts [21, 23], this finding shows that in the best of cases, results similar to those of the null model would be obtained.

## 4.4 Menzerath-Altmann's law fitted parameters

We now fit Eq (2) to empirical data to test whether natural languages and the memoryless source satisfy Menzerath-Altmann's law. The results for Italian, Dutch, Spanish and Portuguese are shown in the panels of Fig 4, while the rest of languages are treated in S2 Fig. For each panel, experimental data is represented with blue circles, the red dashed line is a fit of Eq (2) to the data using a Levenberg–Marquardt algorithm and the memoryless source baseline is represented with the solid gray line. The memoryless source baseline is computed for each language with Eq (4) using the symbol probabilities from Table 2, whereas the fitted parameters of Menzerath-Altmann's law are provided in Table 4 along with the goodness of fit. Following

**Table 3. Menzerath's law tested on the 21 languages under study.** Spearman's rank correlation coefficients and p-values between the size of the word and the mean size of the syllables. The regression function (4) for the memoryless source is monotonically decreasing so Spearman's rank correlation coefficient equals −1 in this case.

| Language | Correlation | p-value |
|---|---|---|
| English | −0.64 | 0.09 |
| French | −1 | $<10^{-2}$ |
| Finnish | 0.14 | 0.64 |
| German | −0.25 | 0.49 |
| Italian | 0.07 | 0.86 |
| Dutch | −0.53 | 0.14 |
| Spanish | −1 | $<10^{-2}$ |
| Portuguese | −0.83 | $<10^{-2}$ |
| Hungarian | −0.68 | 0.04 |
| Swedish | −0.26 | 0.53 |
| Esperanto | −0.96 | $<10^{-2}$ |
| Latin | −0.89 | $<10^{-2}$ |
| Danish | −0.62 | 0.1 |
| Tagalog | −1 | $<10^{-2}$ |
| Catalan | −0.77 | 0.07 |
| Polish | −1 | $<10^{-2}$ |
| Norwegian | −0.61 | 0.15 |
| Czech | −0.83 | 0.04 |
| Welsh | −1 | $<10^{-2}$ |
| Icelandic | 0.0 | 1.0 |
| Afrikaans | −0.83 | 0.04 |

previous studies [12, 15], we have omitted constructs of the smallest size, i.e., monosyllabic words since it has been reported that they do not follow the Menzerath-Altmann's law in several occasions [12]. For completeness, the results including monosyllabic words are addressed in S3 Fig.

Menzerath-Altmann's law is significantly satisfied in almost all considered languages while the memoryless source model is not able to mimic the exponential function of Eq (2) or to exhibit a power law exponent different than −1. Besides, we find that in the majority of languages both $\beta$ and $\gamma$ are less than 0, which leads to a second regime where Menzerath-Altmann's law is inverted [12], as it is markedly appreciated in Fig 4 for the cases of Italian and Dutch. The location of this transition dictated by quotient $\beta/\gamma$ happens approximately for word lengths between 5 to 8 syllables and may be a reflection of complexity and shaping dynamics emerging from lower levels [12]. It is remarkable that while the memoryless source model is able to reproduce Menzerath's law it does not satisfy Menzerath-Altmann's law, suggesting that further research should be focused on the latter.

Finally, we can now explain the values of the correlation coefficients shown in Table 3 in comparison with fitted parameters of Table 4. Languages with high Spearman correlation values as French, Spanish or Welsh do not exhibit the transition point to the second inverted regime of Menzerath-Altmann's law. Instead, there holds a fully monotonic dependency between the parts and the whole. On the other hand, the reason why languages such as English, Finnish, Italian or Dutch do not show a significant negative correlation between the parts and the construct, is actually an artifact caused by the second regime of Menzerath-Altmann's law.

**Table 4. The estimated parameters of MAL for the Gutenberg Corpus.** Fitting of MAL to the experimental data has been done using Levenberg–Marquardt algorithm and excluding words with only one syllable. $R^2$ (coefficient of determination) is used to determine the goodness of the fit. Column $\beta/\gamma$ corresponds to the observable extremum of MAL for $\beta \cdot \gamma > 0$ or is left blank otherwise.

| Language | $\alpha$ | $\beta$ | $\gamma$ | $R^2$ | $\beta/\gamma$ |
|---|---|---|---|---|---|
| English | 3.19 | $-3.5 \cdot 10^{-1}$ | $-5.4 \cdot 10^{-2}$ | 0.82 | 6.5 |
| French | 2.96 | $-2.4 \cdot 10^{-2}$ | $2.6 \cdot 10^{-2}$ | 0.94 | - |
| Finnish | 2.69 | $-5.6 \cdot 10^{-2}$ | $-8.8 \cdot 10^{-3}$ | 0.63 | 6.4 |
| German | 3.19 | $-1.5 \cdot 10^{-1}$ | $-2.4 \cdot 10^{-2}$ | 0.62 | 6.3 |
| Italian | 2.59 | $-1.8 \cdot 10^{-1}$ | $-3.6 \cdot 10^{-2}$ | 0.41 | 5.1 |
| Dutch | 3.27 | $-1.4 \cdot 10^{-1}$ | $-1.3 \cdot 10^{-2}$ | 0.63 | 10.9 |
| Spanish | 2.6 | $-5.3 \cdot 10^{-2}$ | $4.6 \cdot 10^{-3}$ | 0.98 | - |
| Portuguese | 2.66 | $-1.1 \cdot 10^{-1}$ | $-1.0 \cdot 10^{-2}$ | 0.93 | 10.9 |
| Hungarian | 2.69 | $-8.9 \cdot 10^{-2}$ | $-1.1 \cdot 10^{-2}$ | 0.81 | 7.9 |
| Swedish | 2.82 | $-1.0 \cdot 10^{-1}$ | $-2.0 \cdot 10^{-2}$ | 0.71 | 5.1 |
| Esperanto | 2.69 | $-1.4 \cdot 10^{-1}$ | $-2.0 \cdot 10^{-2}$ | 0.95 | 7.1 |
| Latin | 2.82 | $-1.7 \cdot 10^{-1}$ | $-2.1 \cdot 10^{-2}$ | 0.95 | 8.2 |
| Danish | 2.77 | $-5.2 \cdot 10^{-2}$ | $-6.5 \cdot 10^{-3}$ | 0.44 | 8.1 |
| Tagalog | 2.69 | $-1.1 \cdot 10^{-1}$ | $-3.9 \cdot 10^{-3}$ | 0.98 | 28.7 |
| Catalan | 2.83 | $-1.9 \cdot 10^{-1}$ | $-2.5 \cdot 10^{-2}$ | 0.96 | 7.5 |
| Polish | 3.13 | $-1.6 \cdot 10^{-1}$ | $-4.1 \cdot 10^{-3}$ | 0.99 | 39.5 |
| Norwegian | 2.87 | $-1.4 \cdot 10^{-1}$ | $-2.7 \cdot 10^{-2}$ | 0.67 | 5.2 |
| Czech | 2.76 | $-2.3 \cdot 10^{-1}$ | $-3.5 \cdot 10^{-2}$ | 0.95 | 6.7 |
| Welsh | 3.3 | $-2.8 \cdot 10^{-3}$ | $5.9 \cdot 10^{-2}$ | 0.99 | - |
| Icelandic | 2.67 | $5.8 \cdot 10^{-2}$ | $1.9 \cdot 10^{-2}$ | 0.13 | 3.0 |
| Afrikaans | 3.4 | $-3.0 \cdot 10^{-1}$ | $-4.8 \cdot 10^{-2}$ | 0.97 | 6.2 |

## 4.5 Syllable sizes depends on position within word

Whereas the memoryless source is able to reproduce Menzerath's law, it does so at the expense of predicting that all consonant clusters within a word are equally long on average. In this section we would like to show that this prediction is not satisfied by human languages. Namely, the mean size of a syllable strongly depends on its position in the random word. In order to explore the dependence between the mean size of a syllable and its position within a word, we will standardize the position $i = 1, 2, \ldots, m$ of a syllable within an $m$-syllable word using formula

$$x = \begin{cases} \dfrac{i-1}{m-1} & \text{if } m > 1, \\[2mm] 0.5 & \text{if } m = 1. \end{cases} \tag{6}$$

Thus, the standardized position $x$ takes values in range [0, 1]. In the following, we investigate how the mean length of a syllable depends on $x$. The results for Hungarian, Esperanto, Tagalog, and English are depicted in the panels of Fig 5 while the results for 17 remaining languages are provided in S4 Fig. Distinct colours represent different word sizes. Data points, corresponding to distinct values of the standardized position $x$, are joined with solid lines for ease of reading. It is observable that while monosyllabic words seem scattered at random, an order emerges in the plots for longer words. Namely, there are two local extrema: a maximum around the second syllable and a minimum around the penultimate syllable. This behaviour seems universal for the 21 languages under study, and it may reflect some complex dynamics

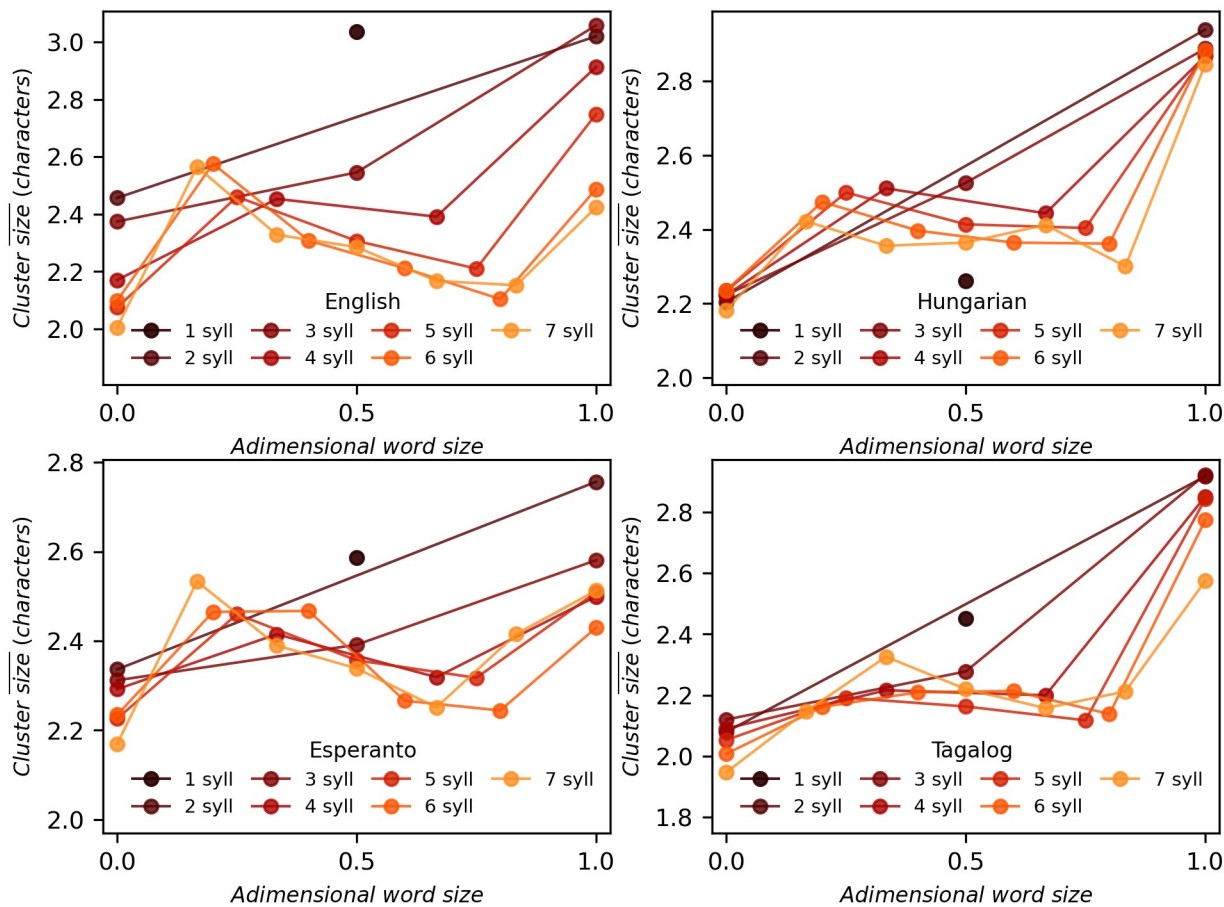

**Fig 5. Syllable sizes depending on word length and position.** The mean size of syllables in the number of characters for different word sizes and syllable positions for the case of English, Hungarian, Esperanto and Tagalog. The position in the word has been standardized according to Eq (6), 0 being the first syllable and 1 being the last syllable. The mean size value for monosyllabic words is represented with a black circle at *x* = 0.5. The results for 17 additional languages are provided in S4 Fig.

that leads to Menzerath-Altmann's law. More comments regarding this phenomenon are provided in the Discussion.

## 5 Discussion

In the previous sections, we have investigated Menzerath's law, which states that the longer a word is the shorter its syllables are on average. In contrast, a more precise mathematical formula for this regression has been called Menzerath-Altmann's law (MAL). Under some circumstances, MAL predicts also an inverted regime, i.e., the syllables get ultimately longer as the words become longer. We have shown that Menzerath's law, but not MAL, arises also for a simple memoryless source with three emitted symbols. This observation shows that the mere Menzerath's law is not a property that distinguishes complex systems, which makes necessary to review and revise some previous comparative studies in human and non-human communication [21–28] or in other fields including genomics or ecology [17, 18, 68]. Moreover, as we have shown analyzing the Standarized Gutenberg Corpus on 21 languages, beyond a certain length of words, the inverted regime is found indeed, providing new evidence for the applicability of MAL. Our results also report a complicated dependency of the average syllable length

depending on its position within the word that may be related with the appearance of MAL and the emergence of the inverted regime. In the following, we discuss in depth the scope and significance of our results.

The first question that arises when studying Menzerath's law and MAL is related to the methodology of computing the mean syllable length. We have observed very similar results for two different methods of computing these averages: first, treating the full corpus as a single text, and second, averaging within each book separately and then reporting the average of averages across all books. On one hand, this observation confirms a certain robustness of MAL, but on the other hand, it raises further questions about the methodology of fitting the MAL formula. We have considered only two different cases, whereas there are many more factors that may—or may not—affect the results, including the lemmatization process or selecting whether MAL is fitted to the average over word tokens or the average over word types. Although the Brevity law ensures that words with similar sizes cluster with similar ranges of frequencies, the self-similarity of Zipf's law ensures that there will be a power law rank-frequency dynamic in the cluster of words with same number of syllables [12], fact that in principle could greatly determine the results. This syllabic structure distribution in each language was recently studied based on the frequency with which symbols occurred, showing an important statistical influence of high frequency words, generally monosyllabic and shorter and that undergo changes earlier than low frequency words, so they tend to show complex syllabic structures [69, 70]. Those and other considerations about the methodology—including also size effects—on the study of MAL have been overlooked but they may affect the conclusions to be drawn from the results. Probably, future works should consider the extensive research and discussions carried out during last years around Zipf's law [66, 71–74], in order to evaluate the robustness of MAL and to establish a standardized methodology that would permit reliable comparative studies.

Subsequently, we would like to discuss implications of deriving an exact formula for Menzerath's law in case of the memoryless source. The memoryless source with three emitted symbols, where symbol occurrences are independent random variables, is the simplest stochastic process imaginable, yet it is able to reproduce Menzerath's law. Thus, the mere Menzerath law cannot be interpreted as a symptom or a reflection of complexity of the signal or of the communication system. The memoryless source model reproduces the negative correlation between the size of the constituents and the size of the whole. Therefore, recent studies that have found Menzerath's law in the communication systems of some animal species and attributed it to a certain *complexity* of communication systems of these living beings [21–28] should be carefully reviewed. The memoryless source model considered in this paper is a baseline model operating in terms of grapheme classes but it could be easily generalized and extended to other types of constituents and substructures such as phonemes, sentences, or breath groups that are typical of speech. In the future work it may be illuminating to study whether some more complicated Markov model can—or cannot—account for some peculiarities of MAL.

Computing the Spearman rank correlation, we found that approximately only a half of the 21 natural languages analyzed presented statistically significant evidence of satisfying Menzerath's law, in opposition to the null model that always obeys it regardless of the parameter values. However, in the second step we fitted the MAL equation to all the languages recovering in all but three cases a second inverted regime. We interpret that this regime is the cause why some languages were not apparently showing any significant negative Spearman rank correlation between the parts and the construct. Without the precaution of fitting the parameters of the mathematical function, erroneous conclusions could be reached, and as it was discussed in previous research [12], this second inverted regime could be a deciding factor whether the system studied is a complex system. In fact, these differences between Menzerath's law and MAL

were related to some debates about the complexity in genomics that took place during previous years [17–19, 30, 75, 76]. For the purpose of comparing with previous studies, we removed monosyllabic words from the analysis, although they are included in S2 Fig. As it was shown previously for English, although MAL is always fulfilled in orality, MAL for written texts or phonetic transcriptions is satisfied only if monosyllables are excluded (see Fig 11 in [12]).

We have finally explored the structure of syllables according to their position in an attempt to better understand the origin of MAL, showing that the syllable size in graphemes strongly depends on its position within the word in quite a universal way for all languages. A feasible hypothesis could be that written texts reproduce the structure of speech phonotactics [69, 77], with three different phases in the word (see Fig 5), caused by the limited capacity of the lungs [78, 79]:

1. Speakers tend to produce longer, more complex syllables at the beginning of the word, so more information is first sent (except for the first syllable or monosyllables, which are usually high frequency words that should be modeled by other variables [70]). That is, that there would be physiological and efficiency pressures to put more information on the first part of the constructs.

2. According to the length of the word, the mean duration of syllables decreases as the pronunciation progresses (compression principle [29]). This generates the tendency that the larger the words are, the smaller the size of their syllables is (Menzerath's law).

3. Towards the end, speakers realize that they can allow themselves to extend the emission until exhausting the pulmonary air, increasing the length and complexity of the final syllable, when possible [12, 69].

Another possible cause of this internal statistical variation in the size in characters of each syllable according to its position was hypothesized by Polikarpov [80] (see page 227). Polikarpov conducted a differentiated analysis of the size in characters of the prefixes, suffixes and lexical roots. In this sense, in addition to the greater complexity and length observed in the syllabic structures at the beginning and at the end of the word, statistical differences due to the type of morpheme could be added.

Actually, it could be argued that MAL, at least as studied here, is a law with a phonetic origin and that therefore a model based on discrete symbols, as is the case of the memoryless source, departs from the continuous reality of speech signals but both written texts and phonetic transcriptions follow MAL [12, 15]. Menzerath himself analyzed the complexity of the continuous chain of speech sounds, defining the phenomenon of coarticulation [46]. Coarticulation demonstrates the complexity in establishing exact limits in the transitions between speech sounds, which has been subsequently verified [35, 36]. Our findings support, therefore, the correspondence between writing and orality. In other words, the distribution of graphemes in written texts would be a reflection of the complexity of speech [12, 78, 79].

To sum up, we have shown that studying MAL correctly implies reviewing the chosen study units to research on the internal correlations within the linguistic units, as well as to compare with null models—with known variations as *intermittent silence*, *monkey typing*, . . . [32, 33, 42], in order to obtain feasible evidence of complexity in communication systems.

## 6 Conclusion

In this paper, we have shown that a simple memoryless source, which is the simplest imaginable stochastic process, is able to reproduce Menzerath's law, questioning its interpretations in terms of communication complexity and efficiency. For this reason comparative studies that

only cover Menzerath's law should be carefully reviewed, especially if they ascribe that this statistical regularity is a consequence of complexity of the involved communication system. In contrast, it seems that this is not the case of the more specific Menzerath-Altmann law, which we have massively explored in texts written in 21 languages. In most of them, we have found an inverted regime that can not be reproduced by the memoryless source model. We think that this inverted regime is the crucial feature that singles out Menzerath-Altmann's law for complex communication systems.

Our analysis, therefore, indicates that the study of Menzerath's law should be focused on fitting and interpretation of specific mathematical functions rather than on exploring shallow correlations, which can be spurious. Furthermore, we have discussed the need for establishing a common methodology to compute the mean syllable length, which is still lacking in the literature, hindering rigorous comparative studies. To make this problem more transparent, we have explicitly compared two distinct ways of computing the mean syllable length and we have reported that they yield very similar results. Finally, wondering how this law emerges, we have explored the average syllable length as depending on the position in the word, finding an unreported dynamics, which may be the cause of Menzerath-Altmann's law at the higher level. Definitely, we consider that future works should investigate the utility of null stochastic models in the study of Menzerath's law and they should also actively seek a standardized methodology to study it.

## 7 Data accessibility

Gutenberg corpus is a freely accessible corpus. Python scripts for downloading, processing the corpus, reproducing results and figures and the numerical results are now available at https://github.com/ivangtorre/Can-Menzerath-law-be-a-criterion-of-complexity-incommunication. We have used Python 3:6:9 for the analysis and Python libraries including nltk 3:5 for the analysis, Scipy 1:5:3, Numpy 1:19:4 Pandas 1:1:4 and Matplotlib 3:3:2.

## Supporting information

**S1 Fig. Menzerath-Almann's law for full corpus vs averaging by book for all studied languages from the standardized Gutenberg project, omitting those presented in the main text.** Gray solid lines indicate the MAL for each book analyzed separately, and the average MAL with black big circles and the solid line is just included for visual representation. In contrast, blue square points represent MAL computed directly for the full corpus treated as a single text. For all cases results are almost the same and there are hardly any differences between both methods.
(PDF)

**S2 Fig. Menzerath-Altmann's law and memoryless source baseline for full corpus excluding monosyllables.** It has been usual in the literature to exclude in the study of MAL elements with only one constituent (in this case words with only one syllable) as it is known that sometimes they do not follow Menzerath's law. However, we here include experimental data and fits for Menzerath-Altmann's law in the standardized Gutenberg corpus for languages not included in the main text, omitting the first point (monosyllables) and comparing with the memoryless source baseline.
(PDF)

**S3 Fig. Menzerath-Altmann's law and memoryless source baseline for full corpus including monosyllables.** It can be seen that, depending on the language, Menzerath's law is satisfied

and the memoryless source model is a good baseline.
(PDF)

**S4 Fig. Syllable sizes depending on word length and position in the word for 17 languages not shown in the main text.** The size of the word has been normalized where 0 corresponds to the first syllable and 1 corresponds to the last syllable. The mean size of monosyllabic words is represented with a black circle in $x = 0.5$.
(PDF)

## Acknowledgments

We want to dedicate this work to the memory of Professor Gabriel Altmann who humbly always encouraged us to investigate "Menzerath's law". *Sit tibi terra levis*, Professor Altmann.

## Author Contributions

**Conceptualization:** Iván G. Torre, Łukasz Dębowski, Antoni Hernández-Fernández.

**Data curation:** Iván G. Torre, Łukasz Dębowski.

**Formal analysis:** Iván G. Torre, Łukasz Dębowski.

**Funding acquisition:** Antoni Hernández-Fernández.

**Investigation:** Iván G. Torre, Łukasz Dębowski, Antoni Hernández-Fernández.

**Methodology:** Iván G. Torre, Łukasz Dębowski, Antoni Hernández-Fernández.

**Project administration:** Łukasz Dębowski, Antoni Hernández-Fernández.

**Resources:** Łukasz Dębowski, Antoni Hernández-Fernández.

**Software:** Iván G. Torre.

**Supervision:** Iván G. Torre, Łukasz Dębowski, Antoni Hernández-Fernández.

**Validation:** Iván G. Torre, Łukasz Dębowski, Antoni Hernández-Fernández.

**Visualization:** Iván G. Torre.

**Writing – original draft:** Iván G. Torre, Łukasz Dębowski, Antoni Hernández-Fernández.

**Writing – review & editing:** Iván G. Torre, Łukasz Dębowski, Antoni Hernández-Fernández.

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
