## [Decision Letter · Decision Letter 0]

30 Jun 2021

PONE-D-21-10246

Can Menzerath's law be a criterion of complexity in communication?

PLOS ONE

Dear Dr. Hernández-Fernández,

Thank you for submitting your manuscript to PLOS ONE. After careful consideration, we feel that it has merit but does not fully meet PLOS ONE’s publication criteria as it currently stands. Therefore, we invite you to submit a revised version of the manuscript that addresses the points raised during the review process.

We look forward to receiving your revised manuscript.

Kind regards,

Diego Raphael Amancio

Academic Editor

PLOS ONE

Journal Requirements:

3. Please upload a copy of Supporting Information which you refer to in your text on pages 16, 18, 19, 21, 22, 23.

Reviewers' comments:

Reviewer's Responses to Questions

**Comments to the Author**

1. Is the manuscript technically sound, and do the data support the conclusions?

Reviewer #1: Yes

Reviewer #2: Yes

2. Has the statistical analysis been performed appropriately and rigorously? 

Reviewer #1: Yes

Reviewer #2: Yes

3. Have the authors made all data underlying the findings in their manuscript fully available?

Reviewer #1: Yes

Reviewer #2: Yes

4. Is the manuscript presented in an intelligible fashion and written in standard English?

Reviewer #1: Yes

Reviewer #2: Yes

5. Review Comments to the Author

Reviewer #1: The manuscript lays out a careful and detailed study of Menzerath's law in the Standardized Project Gutenberg Corpus. As far as I know, this is the most detailed, careful and exhaustive analysis of Menzerath's law ever done.

The authors make a precise distinction between Menzerath's law, which is a more general statement, and Menzerath-Altmann law, which corresponds to a concrete functional form (eq 1). They introduce from the beginning a null model, namely a memoryless process that produces vowels, consonants and spaces. This is justified because it is the simplest stochastic process that makes the law observable. Interestingly, the null model turns out to reproduce Menzerath's law in its standard form, but not the inverted regime.

The analysis comprises books from the Project Gutenberg in 21 languages and takes great care of important questions related to robustness with respect to different methodological choices. In addition, the authors provide access to all code and data used in their analysis, which will allow readers to verify and build upon their work.

Overall, I believe this is an excellent manuscript that should be published in its present form, and is likely to quickly become a standard reference for researchers working on Menzerath's law.

Reviewer #2: The submitted paper answers several questions related to the Menzerath-Altmann law. The law is one of current topics not only in linguistics, but also in other branches of sciences (animal communication and cognition, DNA structure, etc.).

The paper convincingly shows that one should make a careful distinction between the Menzerath law (understood as a correlation between the sizes of the whole and its parts) and the Menzerath-Altmann law (a mathematically formulated expectation on the mean length of parts). The mathematical and statistical apparatus used is sound and I do not have any comments in this respect.

Minor remarks:

1) It seems that the authors use word tokens, as opposed to word types. Their choice should be explicitly stated in the paper (as it makes a difference). See eg https://www.degruyter.com/document/doi/10.1515/lingvan-2019-0076/html , where the relation between word and morpheme length is studied, and the difference between the choice of types vs. tokens is addressed.

2) p. 2, line 30, replace "an spurious" with "a spurious"

3) p. 2, line 43

The text as it is written in the current version of the paper seems to admit also alpha<0. It would be better to reformulate it, eg "being alpha>0, and usually beta<0 and gamma <0", or something similar.

4) p.76, lines 76 and 77

While orality certainly preceded written texts, eg in syntax the structure in written texts does not probably to have to reflect speech (eg very long sentences preferred by some novelists were probably never meant to be pronounced). The statement from the paper could be true for the level of word and syllable studied in the submitted paper, but maybe not in general.

5) p. 5, line 109

In fact, there are at least two papers which use simulations in the context of the MAL research:

https://www.degruyter.com/document/doi/10.1515/9783110362879-005/html

https://www.degruyter.com/document/doi/10.1515/9783110362879-009/html

6) p. 8, lines 156-158

Diphthongs and triphthongs are single vowels. Of course the choice to consider graphemes (as opposed to phonemes) indeed vbrings the problem which the authors discuss on these lines, but the text should be reformulated (eg "the vowel which is the syllable nucleus can be written by two or three graphemes in case of diphthons and triphthongs", or something similar).

7) p.10, Table 1

The graphemes acting as different phonemes were assigned to the most frequent class - this is true for each language separately, all across all languages used in the study?

8) p. 11, lines 208, 209

"both the onset and the coda have a decreasing sonority"

The onset has an increasing sonority - or decreasing, if one moves from the nucleus towards the beginning of the syllable. The text should be made more precise.

9) p.12, line 228

replace "regression 4" with "regression (4)"

10) p.21, lines 308-310

If word tokens (as opposed to word types) were taken, it should be explicitly mentioned at this place as well. If fact, the supposed irregular behaviour of monosyllables can be - for word tokens - explained by the Zipf law. Monosyllables occur on average more often than longer words (Zipf), the syllable (the word itself in this case) is on average long (Menzerath), and within monosyllables shorter ones are preferred (Zipf again). This interaction of the two laws can lead to the irregularities. they should diappear of word types are considered.

11) p.27, line 435

Polikarpov is correctly cited here, but Grzybek (p. 36, lines 651, 652) appears in the references as [76] - Grzybek is the editor of the volume in which Polikarpov's paper appeared.

12) p. 27, line 443

A law with a phonetic orogin - probably yes, at this level (word-syllable); this statement can be more doubtful eg for syntacotc structure, semantic aggregates introdced by Hrebicek (for which the MAL is laso valid), etc.

6. PLOS authors have the option to publish the peer review history of their article (what does this mean?). If published, this will include your full peer review and any attached files.

Reviewer #1: No

Reviewer #2: No

---

## [Author Response · Author response to Decision Letter 0]

7 Jul 2021

Editor:

We have now changed the name of the files according to the required format and also addressed some other small changes on Table and Figure names.

Thank you very much for this comment. We ensure now that we provide the correct grant numbers for the grants received for this study in the ‘Funding Information’ section and in Financial Disclosure section. 

3. Please upload a copy of Supporting Information which you refer to in your text on pages 16, 18, 19, 21, 22, 23.

We don’t know if this is a technical error in the first submission but anyway we resend the supplementary information again. We are sorry if there has been any problem in this regard.

Reviewer 1.

We really appreciate the comments and analysis of Reviewer 1. We are glad that she/he liked the work.

Reviewer 2.

We thank reviewer 2 for the interesting suggestions that have led to an improvement of the final version of the article. In the following we answer them point by point.

1) It seems that the authors use word tokens, as opposed to word types. Their choice should be explicitly stated in the paper (as it makes a difference). See eg https://www.degruyter.com/document/doi/10.1515/lingvan-2019-0076/html , where the relation between word and morpheme length is studied, and the difference between the choice of types vs. tokens is addressed.

We appreciate this comment and we have made some small changes pointing out that we have used word tokens instead of word types during the research.

Abstract:

“In this paper, we investigate the anatomy of MAL for constructs being word tokens and constituents being syllables, measuring its length in graphemes.”

Actual line 85:

“In this paper, we will adopt word tokens, syllables and graphemes as the linguistic units to massively study...”

Actual line 185:

“... and encompassing more than 10000 books and roughly 6x10^8 word tokens.”

Table2:

Second column is now “Tokens”.

Table 2 Caption: 

“Summary of characteristics for the 21 languages studied including the total number of books, word tokens, syllables….”

Line 253:

“Table 2 summarizes, for each language, the main characteristics of the corpus, including total number of books, word tokens …”

2) p. 2, line 30, replace "an spurious" with "a spurious"

Fixed.

3) p. 2, line 43. The text as it is written in the current version of the paper seems to admit also alpha<0. It would be better to reformulate it, eg "being alpha>0, and usually beta<0 and gamma <0", or something similar.

Thanks for the suggestion, we have fixed it.

4) p.76, lines 76 and 77. While orality certainly preceded written texts, eg in syntax the structure in written texts does not probably have to reflect speech (eg very long sentences preferred by some novelists were probably never meant to be pronounced). The statement from the paper could be true for the level of word and syllable studied in the submitted paper, but maybe not in general.

We agree with the reviewer and we have been more general in the statement. 

Previous one (lines 76-77): “It is worth addressing that linguistic structures present in written texts are probably a reflection of orality.”

New: “It is worth addressing that linguistic structures present in written texts are probably a reflection of orality, although some complex structures in writing may be rare in orality.”

5) p. 5, line 109 In fact, there are at least two papers which use simulations in the context of the MAL research:

https://www.degruyter.com/document/doi/10.1515/9783110362879-005/html

https://www.degruyter.com/document/doi/10.1515/9783110362879-009/html

Thank you very much for these valuable references. We have included these references and one more recent work by Kumiko Tanaka-Ishii, published in Entropy in May, 2021. In addition to this we have slightly modified and extended the text (lines 112-117):

Random models for Menzerath-Altmann's law were previously discussed in Linguistics at the sentence-clause level considering several randomized versions of the original text [benevsova2015], and also for word length motifs on an extensive Modern Greek corpus [mavcutek2015], where the inverted regime in MAL was not considered by the random model (see figure 1 in [mavcutek2015]). Recently, Menzerath's law has been recovered in the syntax of several languages with random sentences, determining with some approximation that this law can be reproduced at the syntactic level by randomization [Tanaka2021].

6) p. 8, lines 156-158. Diphthongs and triphthongs are single vowels. Of course the choice to consider graphemes (as opposed to phonemes) indeed brings the problem which the authors discuss on these lines, but the text should be reformulated (eg "the vowel which is the syllable nucleus can be written by two or three graphemes in case of diphthons and triphthongs", or something similar).

We appreciate this comment. Now the sentence is:

“However, in natural language, the syllabic nucleus can also be written by two or three graphemes yielding diphthongs or triphthongs”

7) p.10, Table 1. The graphemes acting as different phonemes were assigned to the most frequent class - this is true for each language separately, all across all languages used in the study?

Thank you very much for this comment. In a first approximation, the classification has been made taking into account Spanish, English and Polish. In a second approach, other languages have been considered as new graphemes appeared in the preliminary analysis. Since the sentence that appeared in the text was not entirely rigorous, we have chosen to remove it.

8) p. 11, lines 208, 209. "both the onset and the coda have a decreasing sonority". The onset has an increasing sonority - or decreasing, if one moves from the nucleus towards the beginning of the syllable. The text should be made more precise.

We agree with the reviewer and we change this sentence (lines 208-210) “SSP states that the nucleus of the syllable is the element of the maximum sonority, so that both the onset and the coda have a decreasing sonority.” and finally write more precisely:

“SSP states that the nucleus of the syllable is the element of the maximum sonority. Although some linguistic varieties of languages not studied here (Russian, Arabic) may not strictly follow this principle and various formulations of SSP have been discussed, the SSP states that only sounds of higher sonority rank are permitted between any element of a syllable and the syllable peak, considering that clusters violating this principle do occur but they are infrequent (\\cite{clements1990role}, p.285).”\\\\

Reference: Clements, G. N. (1990). The role of the sonority cycle in core syllabification. In J. Kingston and M. E. Beckman (eds.) Papers in Laboratory Phonology I: Between the grammar and the physics of speech. Cambridge: Cambridge University Press. 283-333.

9) p.12, line 228. replace "regression 4" with "regression (4)"

Fixed.

10) p.21, lines 308-310

If word tokens (as opposed to word types) were taken, it should be explicitly mentioned at this place as well. If fact, the supposed irregular behaviour of monosyllables can be - for word tokens - explained by the Zipf law. Monosyllables occur on average more often than longer words (Zipf), the syllable (the word itself in this case) is on average long (Menzerath), and within monosyllables shorter ones are preferred (Zipf again). This interaction of the two laws can lead to the irregularities. they should diappear of word types are considered.

We considered that with the new changes we have previously detailed that we are using word tokens instead of types. It is true that there are interactions that should be studied between Zipf law and Menzerath Altmann law. We consider that the closest approach to orality is to take into account the tokens since in spoken communication there are differences, for example, in the units of duration or energy between different realizations of tokens for the same type. Besides, those interactions between Zipf and Menzerath should appear similarly in non-monosyllabic words and we do not expect a different behavior between them. In any case this point is beyond the scope of this research paper, but considered in the discussion. Actual lines 372-381:

“We have considered only two different cases, whereas there are many more factors that may—or may not—affect the results, including the lemmatization process or selecting whether MAL is fitted to the average over word tokens or the average over word types. Although the Brevity law ensures that words with similar sizes cluster with similar ranges of frequencies, the self-similarity of Zipf’s law ensures that there will be a power law rank-frequency dynamic in the cluster of words with same number of syllables [12], fact that in principle could greatly determine the results. This syllabic structure distribution in each language was recently studied based on the frequency with which symbols occurred, showing an important statistical influence of high frequency words, generally monosyllabic and shorter and that undergo changes earlier than low frequency words, so they tend to show complex syllabic structures [67,68].”

11) p.27, line 435 Polikarpov is correctly cited here, but Grzybek (p. 36, lines 651, 652) appears in the references as [76] - Grzybek is the editor of the volume in which Polikarpov's paper appeared.

Thanks. We have now correctly cited Polikarpov: “Polikarpov AA. Towards the foundations of Menzerath’s law. In: Contributions to the Science of Text and Language. Springer; 2007. p. 215–240”.

12) p. 27, line 443. A law with a phonetic origin - probably yes, at this level (word-syllable); this statement can be more doubtful eg for syntactic structure, semantic aggregates introduced by Hrebicek (for which the MAL is also valid), etc.

We appreciate this comment and we have slightly changed the text to: “Actually, it could be argued that MAL, at least as studied here, is a law with a phonetic origin and …” However we consider this an interesting discussión and an origin for future work, new theories and explanations about this law.

---

## [Decision Letter · Decision Letter 1]

30 Jul 2021

Can Menzerath's law be a criterion of complexity in communication?

PONE-D-21-10246R1

Dear Dr. Hernández-Fernández,

We’re pleased to inform you that your manuscript has been judged scientifically suitable for publication and will be formally accepted for publication once it meets all outstanding technical requirements.

Kind regards,

Diego Raphael Amancio

Academic Editor

PLOS ONE

Additional Editor Comments (optional):

Reviewers' comments:

Reviewer's Responses to Questions

**Comments to the Author**

1. If the authors have adequately addressed your comments raised in a previous round of review and you feel that this manuscript is now acceptable for publication, you may indicate that here to bypass the “Comments to the Author” section, enter your conflict of interest statement in the “Confidential to Editor” section, and submit your "Accept" recommendation.

Reviewer #2: All comments have been addressed

2. Is the manuscript technically sound, and do the data support the conclusions?

Reviewer #2: Yes

3. Has the statistical analysis been performed appropriately and rigorously? 

Reviewer #2: Yes

4. Have the authors made all data underlying the findings in their manuscript fully available?

Reviewer #2: Yes

5. Is the manuscript presented in an intelligible fashion and written in standard English?

Reviewer #2: Yes

6. Review Comments to the Author

Reviewer #2: All my comments and corrections were satisfactorily addressed. In my opinion, the revised version of the paper can be published.

7. PLOS authors have the option to publish the peer review history of their article (what does this mean?). If published, this will include your full peer review and any attached files.

Reviewer #2: No

---

## [Editor Report · Acceptance letter]

10 Aug 2021

PONE-D-21-10246R1 

Can Menzerath’s law be a criterion of complexity in communication? 

Dear Dr. Hernández-Fernández:

I'm pleased to inform you that your manuscript has been deemed suitable for publication in PLOS ONE. Congratulations! Your manuscript is now with our production department. 

Kind regards, 

on behalf of

Dr. Diego Raphael Amancio 

Academic Editor

PLOS ONE